# A Hybrid Method Based on Extreme Learning Machine and Wavelet Transform Denoising for Stock Prediction

**DOI:** 10.3390/e23040440

**Published:** 2021-04-09

**Authors:** Dingming Wu, Xiaolong Wang, Shaocong Wu

**Affiliations:** College of Computer Science and Technology, Harbin Institute of Technology, Shenzhen 518055, China; hitsz_wudingming@outlook.com (D.W.); wushaocong2013@gmail.com (S.W.)

**Keywords:** stock prediction, extreme learning machine, wavelet transform, deep learning

## Abstract

The trend prediction of the stock is a main challenge. Accidental factors often lead to short-term sharp fluctuations in stock markets, deviating from the original normal trend. The short-term fluctuation of stock price has high noise, which is not conducive to the prediction of stock trends. Therefore, we used discrete wavelet transform (DWT)-based denoising to denoise stock data. Denoising the stock data assisted us to eliminate the influences of short-term random events on the continuous trend of the stock. The denoised data showed more stable trend characteristics and smoothness. Extreme learning machine (ELM) is one of the effective training algorithms for fully connected single-hidden-layer feedforward neural networks (SLFNs), which possesses the advantages of fast convergence, unique results, and it does not converge to a local minimum. Therefore, this paper proposed a combination of ELM- and DWT-based denoising to predict the trend of stocks. The proposed method was used to predict the trend of 400 stocks in China. The prediction results of the proposed method are a good proof of the efficacy of DWT-based denoising for stock trends, and showed an excellent performance compared to 12 machine learning algorithms (e.g., recurrent neural network (RNN) and long short-term memory (LSTM)).

## 1. Introduction

In the era of big data, deep learning for predicting stock prices [1] and trends has become more popular [2,3]. The fully-connected feedforward neural network (FNN) possesses excellent performance, and is superior to traditional time-series forecasting techniques (e.g., autoregressive integrated moving average (ARIMA)) [4,5]. The FNN is mainly trained using the well-known backpropagation (BP) algorithm [6]. The traditional BP algorithm essentially optimizes parameters based on the gradient descent method, accompanied by the problems of slow convergence and local minima. Extreme learning machine (ELM) is one of the effective training algorithms for single-hidden-layer feedforward neural networks (SLFNs). In ELM, hidden nodes are initialized randomly, and network parameters at the input end are fixed without iterative tuning. The only parameter that needs to be learned is the connection (or weight) matrix between the hidden layer and the output layer [7]. Theoretically, if hidden nodes are randomly generated, ELM maintains the general approximation capability of SLFNs [8]. Compared with the traditional FNN algorithm, the advantages of ELM in terms of efficiency and generalization performance have been proven on a wide range of issues in different fields [9]. The ELM possesses a faster learning and better generalization performance [10,11,12] than traditional gradient-based learning algorithms [13]. Due to efficient learning ability of ELM, it is widely used in classification, regression problems, etc. [14,15]. In addition to being used for traditional classification and regression tasks, ELM has recently been extended for clustering, feature selection, and representation learning [16]. For more research on ELM, please refer to related literatures [17,18,19,20,21,22,23,24,25].

In recent years, the research of hybrid models for time series prediction has become more popular [26]. Regarding the advantages of ELM, in recent years, the use of ELM for time-series datasets has gradually increased. A number of scholars have applied ELM to carry out feature engineering on time-series data [27], and ELM was extensively utilized to study various hybrid models for predicting time-series data. In order to discover the features of the original data, Yang et al. proposed an ELM-based recognition framework to deal with the recognition problem [28]. Li et al. proposed the design and architecture of a trading signal mining platform that uses ELM to simultaneously predict stock prices based on market news and stock price datasets [29]. Wang et al. introduced a new mutual information-based sentiment analysis method and employed ELM to improve the prediction accuracy and accelerate the prediction performance of the proposed model [30]. Jiang et al. combined empirical mode decomposition, ELM, and improved harmony search algorithm to establish a two-stage ensemble model for stock price prediction [31]. Tang et al. optimized the ELM by the differential evolution algorithm to construct a new hybrid predictive model [32]. Weng et al. presented an improved genetic algorithm regularized online ELM to predict gold price [33]. Jiang et al. proposed a hybrid approach consisting of pigeon-inspired optimization (PIO) and ELM based on wavelet packet analysis (WPA) for predicting bulk commodity prices. That hybrid model possessed a better performance on horizontal precision, directional precision and robustness [34]. Khuwaja et al. presented a framework to predict the stock price movement using phase space reconstruction (PSR) and ELM, and results achieved from the proposed framework were compared with the conventional machine learning pipeline, as well as the baseline methods [35]. Jeyakarthic and Punitha introduced a new method based on multi-core ELM for forecasting stock market returns [36]. Xu et al. presented a new carbon price prediction model using time series complex network analysis and ELM [37]. In addition to the use of ELM for feature processing and developing hybrid models, a number of studies have modified ELM to make it highly appropriate for a variety of practical scenarios. Wang et al. introduced a non-convex loss function, developed a robust regularized ELM, and emphasized on solving the key problem of low efficiency [38]. Guo et al. presented a robust adaptive online sequential ELM-based algorithm for online modeling and prediction of non-stationary data streams with outliers [39].

The research of stock prediction is inseparable from the hypothesis of market efficiency [40]. There are a lot of studies on market efficiency [41]. The general conclusion is that the market in which the stock trend can be predicted should be the ineffective market, otherwise it is impossible to predict the trend of stock. Relevant studies have pointed out that China’s stock market is a relatively emerging market [42,43], which is relatively ineffective. Therefore, it is feasible to use various machine learning models to predict the trend of the stock through transaction data analysis. In order to predict the stock trend in an ineffective market, we need to consider the influence of noise on trend prediction. Due to the influence of various accidental factors on the financial market, the impact of noise on financial time-series data draws scholars’ attention. To our knowledge, noise often distorts investors’ judgments on stock trends and seriously affects further analysis and processing. However, financial time-series data possess non-stationary and non-linear characteristics, and the traditional denoising processing methods are often accompanied with several defects. The traditional methods of denoising the financial time-series data mainly include the moving average (MA) [44], Kalman filter [45], Wiener filter [46], and fast Fourier transform (FFT) [47]. As a simple data smoothing technique, the MA approximately processes the time-series data. In the denoising process, it may also lead to a loss of useful information. The FFT generally treats high-frequency signals as noise, sets all Fourier coefficients above a certain threshold frequency to zero, and then converts the datasets to the time-domain through inverse Fourier transform to achieve denoising [47]. With respect to the low signal-to-noise ratio in financial data, it is difficult to achieve effective denoising due to more high-frequency effective signals. Kalman filter is a recursive estimator to estimate the state of a system based on the criterion of minimum mean-square error of the residual [48]. As financial time-series data are non-stationary and nonlinear, it is difficult to describe their state and behavior with a definite equation. In signal processing, the Wiener filter is a filter used to produce an estimate of a desired or target random process by linear time-invariant (LTI) filtering of an observed noisy process, assuming known stationary signal and noise spectra, and additive noise. The Wiener filter minimizes the mean square error between the estimated random process and the desired process. Wavelet analysis is a mathematical technique that can decompose a signal into multiple lower resolution levels by controlling the scaling and shifting factors of a single wavelet function. FFT has no locality, while wavelet transform not only has locality, but also scaled parameters that can change the spectrum structure and the shape of the window, so that wavelet analysis can achieve the effect of multi-resolution analysis [49]. The wavelet methods can be used to decompose a noisy signal into different scales and remove the noise while preserving the signal, regardless of the frequency content. The wavelet transforms are developed according to the requirements of time-frequency localization. They possess adaptive properties and are particularly appropriate for processing of stationary and non-linear signals [50]. A recent study employed wavelet transform to reduce the amount of noisy data [51]. Xu et al. proposed a novel method based on the wavelet-denoising algorithm and multiple echo state networks to improve the prediction accuracy of noisy multivariate time-series [52]. Bao et al. developed a deep learning framework, combining wavelet transform, stacked autoencoders, and long short-term memory (LSTM) for stock price prediction [53]. Yang et al. presented an image processing method based on wavelet transform for big data analysis of historical data of individual stocks to obtain images of individual stocks volatility data [54]. Lahmiri introduced a new method based on the combination of stationary wavelet transform and Tsallis entropy for empirical analysis of the return series [55]. Li and Tang proposed a WT-FCD-MLGRU model, which is the combination of wavelet transform, filter cycle decomposition and multi-lag neural networks, and that model possessed minimum forecasting error in stock index prediction [56]. Wen et al. used wavelet transform to extract the features of the Shanghai Composite Index and S&P Index to study the relationship between China’s stock market and international commodities [57]. Mohammed et al. employed continuous wavelet transform and wavelet coherence method to study the relationship between stock indices in different markets [58]. Yang applied the differential method to highlight the trend of the stock price change. To further suppress the influence of stock noise data, they employed wavelet transform to decompose the stock data into principal component and detailed component [59]. Xu et al. presented design and implementation of a stacked system to predict the stock price. Their model used the wavelet transform technique to reduce the noise of market data, and stacked auto-encoder to filter unimportant features from preprocessed data [60]. He et al. proposed a new shrinkage (threshold) function to improve the performance of wavelet shrinkage denoising [61]. Yu et al. used wavelet coherence and wavelet phase difference based on continuous wavelet transform to quantitatively analyze the correlation effect of stock signal returns in the time-frequency domain [62]. Faraz and Khaloozadeh applied wavelet transform to reduce the noise in the stock index and to make the data smooth [63,64]. Chen utilized a recursive adaptive separation algorithm based on discrete wavelet transform (DWT) to denoise data [65]. Li et al. proposed a hybrid model based on wavelet transform denoising, ELM, and k-nearest neighbor regression optimization for stock prediction [66]. There are several other similar studies as well [67,68,69,70,71,72,73,74,75]. Basically, related research has improved the effect of related time-series prediction through wavelet transform.

Due to the applicability of wavelet transform in financial data, as well as the need for data smoothing of the labeling method based on the continuous trend of stock data, and with respect to the advantages of ELM, we proposed a hybrid method for stock trend prediction based on ELM and wavelet transform.

The main arrangements of this paper are as follows:

In the second section, the theories related to ELM and wavelet transform are introduced, and then, the hybrid method DELM (the combined model of wavelet transform denoising and ELM) is described.

In the third section, an overview of the continuous trend labeling method based on time-series data is presented, and the stock datasets used in this paper are introduced. The statistical metrics used for evaluation of experimental results are described, and the reasons for choosing those metrics are discussed.

In the fourth section, the differences between the denoised data and the raw data are compared. Besides, the stationarity testing of the denoised data after feature preprocessing is carried out. The difference in the labeling effect after the combination of the labeling method and DWT-based denoising is analyzed. The prediction results of ELM and DELM are compared. The prediction results of the DELM and other commonly used algorithms are compared.

The descriptions of the related abbreviations are listed in Appendix B
Table A1.

## 2. Methods

In this section, we briefly introduce ELM and wavelet transform theoretically, and describe the DELM method proposed.

### 2.1. ELM

ELM is a new algorithm developed for SLFNs [76]. In an ELM algorithm, the input layer weights are randomly assigned, and the output layer weights are obtained by using the generalized inverse of the hidden layer output matrix. Compared with traditional feed-forward neural network training algorithms, which are slow, easy to fall into local minimum solutions, and are sensitive to the selection of learning rates, the ELM algorithm randomly generates the connection weights of the input layer and the threshold of the hidden layer neurons, and it is unnecessary to adjust the weights in the training process. It is only by setting the number of neurons in the hidden layer, that a unique optimal solution can be obtained. Compared with the previous traditional training algorithms, the ELM algorithm is advantaged by fast learning and good generalization performance. It is not only appropriate for regression and fitting problems, but also for classification [77,78] and pattern recognition. The structure of a typical SLFN is shown in Figure 1. It consists of an input layer, a hidden layer, and an output layer, which are fully connected.

There are *n* neurons in the input layer, corresponding to the number of *n* input variables, and *l* neurons in the hidden layer; there are *m* neurons in the output layer, corresponding to the number of *m* output variables. *W* denotes the weight matrix linking the input layer and the hidden layer, where *w_ij_* is the weight of the *i*-th neuron in the hidden layer and the *j*-th neuron in the input layer. The weight matrix linking the hidden layer and the output layer is *β*, *β_jk_* represents the weight matrix connecting the *j*-th neuron in the hidden layer and the *k*-th neuron in the output layer, and *b* represents the threshold of neurons in the hidden layer. *W*, *β,* and *b* are shown in Equation (1).
(1)$$W={\left[\begin{array}{c} w_{11}\;\;\;w_{12}\;\;\ldots \;\;w_{1n}\\ w_{21}\;\;\;w_{22}\;\;\ldots \;\;w_{2n}\\ \;\ldots \;\;\;\;\;\ldots \;\;\;\ldots \;\;\;\ldots \;\;\\ w_{l1}\;\;\;w_{l2}\;\;\;\ldots \;\;w_{ln} \end{array}\right]}_{l\times n},\;\beta ={\left[\begin{array}{c} \beta_{11}\;\;\;\beta_{12}\;\;\ldots \;\;\beta_{1m}\\ \beta_{21}\;\;\;\beta_{22}\;\;\ldots \;\;\beta_{2m}\\ \;\ldots \;\;\;\;\;\ldots \;\;\;\ldots \;\;\;\ldots \;\;\\ \beta_{l1}\;\;\;\beta_{l2}\;\;\;\ldots \;\;\beta_{lm} \end{array}\right]}_{l\times m},\;b={\left[\begin{array}{c} b_{1}\\ b_{2}\\ \vdots \;\\ b_{l} \end{array}\right]}_{l\times 1}$$

For the training set of *N* samples, the input matrix is *X*, the output matrix is *Y*, and *T* is the expected output matrix (see Equation (2)).
(2)$$\begin{array}{l} X={\left[\begin{array}{c} x_{11}\;\;\;x_{12}\;\;\ldots \;\;x_{1N}\\ x_{21}\;\;\;x_{22}\;\;\ldots \;\;x_{2N}\\ \;\ldots \;\;\;\;\;\ldots \;\;\;\ldots \;\;\;\ldots \;\;\\ x_{n1}\;\;\;x_{n2}\;\;\;\ldots \;\;x_{lN} \end{array}\right]}_{n\times N},Y={\left[\begin{array}{c} y_{11}\;\;\;y_{12}\;\;\ldots \;\;y_{1N}\\ y_{21}\;\;\;y_{22}\;\;\ldots \;\;y_{2N}\\ \;\ldots \;\;\;\;\;\ldots \;\;\;\ldots \;\;\;\ldots \;\;\\ y_{m1}\;\;\;y_{m2}\;\;\;\ldots \;\;y_{mN} \end{array}\right]}_{m\times N}\\ T={\left[\begin{array}{c} t_{11}\;\;\;t_{12}\;\;\ldots \;\;t_{1N}\\ t_{21}\;\;\;t_{22}\;\;\ldots \;\;t_{2N}\\ \;\ldots \;\;\;\;\;\ldots \;\;\;\ldots \;\;\;\ldots \;\;\\ t_{m1}\;\;\;t_{m2}\;\;\;\ldots \;\;t_{mN} \end{array}\right]}_{m\times N} \end{array}$$

The goal of ELM for learning SLFN is to minimize the output error, which is expressed in Equation (3).
(3)$$\sum_{j=1}^{N}\left\|y_{j}-t_{j}^{}\right\|=0,\;t_{j}={\left[t_{1j},t_{2j},\ldots ,t_{mj}\right]}^{\prime },\;y_{j}={\left[y_{1j},y_{2j},\ldots ,y_{mj}\right]}^{\prime }$$

That is, the existence of *w_i_*, *β_i_* and *b_i_* results to hold Equation (4). It can therefore be expressed as Equation (5).
(4)$$\sum_{i=1}^{l}\beta_{i}g\left(w_{i}\cdot x_{j}+b_{i}\right)=t_{j},j=1,\ldots ,N$$
(5)Hβ=T

With expanding Equations (4) and (5) to Equations (6) and (7), we can achieve the specific network output form, as well as the specific form of *H*.
(6)$$\begin{array}{l} t_{j}=\left[\begin{array}{c} t_{1j}\\ t_{2j}\\ \vdots \\ t_{mj} \end{array}\right]=\left[\begin{array}{c} \sum_{i=1}^{l}\beta_{i1}g\left(w_{i}x_{j}+b_{i}\right)\\ \sum_{i=1}^{l}\beta_{i2}g\left(w_{i}x_{j}+b_{i}\right)\\ \vdots \\ \sum_{i=1}^{l}\beta_{im}g\left(w_{i}x_{j}+b_{i}\right) \end{array}\right]\;\;\;\;(j=1,2,\ldots N),\;\\ w_{i}=\left[w_{i1},w_{i2},\ldots ,w_{in}\right],\;x_{j}={\left[x_{1j},x_{2j},\ldots ,x_{nj}\right]}^{\prime } \end{array}$$
(7)$$\begin{array}{l} H\left(w_{1},w_{2},\ldots ,w_{l},b_{1},b_{2},\ldots ,b_{l},x_{1},x_{2},\ldots ,x_{N}\right)=\;\;\\ \begin{array}{l} {\left[\begin{array}{c} g\left(w_{1}\cdot x_{1}+b_{1}\right)\;\;\;\;\;g\left(w_{2}\cdot x_{1}+b_{2}\right)\;\;\;\;\cdots \;\;\;\;g\left(w_{l}\cdot x_{1}+b_{l}\right)\\ g\left(w_{1}\cdot x_{2}+b_{1}\right)\;\;\;\;\;g\left(w_{2}\cdot x_{2}+b_{2}\right)\;\;\;\;\cdots \;\;\;\;g\left(w_{l}\cdot x_{2}+b_{l}\right)\\ \;\cdots \;\;\;\;\;\;\;\;\;\;\;\;\;\;\;\;\;\;\;\;\;\cdots \;\;\;\;\;\;\;\;\;\;\;\cdots \;\;\;\;\;\;\;\;\;\;\;\cdots \\ g\left(w_{1}\cdot x_{N}+b_{1}\right)\;\;\;\;g\left(w_{2}\cdot x_{N}+b_{2}\right)\;\;\;\;\cdots \;\;\;\;g\left(w_{l}\cdot x_{N}+b_{l}\right) \end{array}\right]}_{N\times l}\;\; \end{array} \end{array}$$

In order to train a SLFN, we attempted to obtain ${\hat{w}}_{i}$, ${\hat{b}}_{i}$ and ${\hat{\beta }}_{i}$, resulting in holding Equation (8). This minimized the loss function in Equation (9).
(8)$$\left\|H\left({\hat{w}}_{i},{\hat{b}}_{i}\right){\hat{\beta }}_{i}-T\right\|=\underset{w,b,\beta }{\min}\left\|H\left(w_{i},b_{i}\right)\beta_{i}-T\right\|,i=1,\ldots ,l$$
(9)$$E={\sum_{j=1}^{N}\left(\sum_{i=1}^{l}\beta_{i}g\left(w_{i}\cdot x_{j}+b_{i}\right)-t_{j}\right)}^{2}$$

Some traditional algorithms based on gradient descent can be used to solve such problems, while it is necessary to adjust all parameters in an iterative process for gradient-based learning algorithms. For the ELM algorithm, once the input weights and the bias of the hidden layer are randomly determined, the output matrix of the hidden layer is uniquely constructed. The output weights can be determined by the system (i.e., the ELM algorithm randomly assigns input weights and hidden layer bias rather than completely adjusting all parameters (e.g., backpropagation neural network (BPNN)). For SLFNs, the ELM algorithm can analytically determine output weights. Through the proof of the previously presented theorems Theorems 2.1 and 2.2 in [76], the minimum norm of the weights is given, which is simple to implement and fast in prediction. Therefore, *W* and *b* are randomly selected and determined, and remain unchanged during the training process. *β* can be obtained by solving the least squares of the Equations (10) and (11).
(10)$$\underset{\beta }{\min}\left\|H\beta -T\right\|$$
(11)$$\hat{\beta }=H^{+}T$$
where, *H*^+^ is the Moore–Penrose generalized inverse of the hidden layer output matrix *H*.

### 2.2. Wavelet Analysis

Wavelet transform is a mathematical approach that gives the time-frequency representation of a signal with the possibility to adjust the time-frequency resolution. It can simultaneously display functions and manifest their local characteristics in time-frequency domain. The use of these characteristics facilitates training of neural networks with accuracy to extremely nonlinear signals. It is a time-frequency localized analysis method that the size of window is fixed, while its shape may change [79]. In other words, it has a lower time resolution and higher frequency resolution in low frequency band [80], and higher time resolution and lower frequency resolution in high frequency band, making it highly appropriate for analyzing non-stationary signals, as well as extracting the local characteristics of signals. Wavelet transform includes continuous wavelet transform (CWT) and DWT [81]. The aim of wavelet transform is to translate a function called basic wavelet with parameter *τ*, then, scale the function with the scaling parameter *a*, and do inner product with signal *x*(*t*), as formulated in Equation (12), where *a* > 0 is the scaling factor used to scale the *φ*(*t*) function, and the *τ* parameter is used to translate the function *φ*(*t*). Both *a* and *τ* are continuous variables, thus, Equation (12) is called CWT [82]. DWT is a transform that decomposes a given signal into a number of sets, where each set is a time-series of coefficients describing the time evolution of the signal in the corresponding frequency band [83]. DWT discretizes a signal according to the power series based on the scaling parameter, and is often used for signal decomposition and reconstruction [84]. DWT constructs a scaling function vector group and a wavelet function vector group at different scales and time periods, i.e., the scaling function vector space and the wavelet function vector space [85]. At a certain level, the signal convolved in the scaling space is the approximated, low-frequency information of the raw signal (e.g., “cA” component in Figure 2), and the signal convolved in the wavelet space is the detailed, high-frequency information of the raw signal (e.g., “cD” component in Figure 2). DWT has two important functions, one is the scaling function, as shown in Equation (13), and the other is the wavelet function, as presented in Equation (14) [79].
(12)$$WT_{x}\left(a,\tau \right)=\frac{1}{\sqrt{a}}\int_{-\infty }^{+\infty }x\left(t\right)\varphi \left(\frac{t-\tau }{a}\right)dt$$
(13)$$\phi_{jk}\left(t\right)=2^{-\frac{j}{2}}\phi \left(2^{-j}t-k\right),j,k\in Z$$
(14)$$\psi_{jk}\left(t\right)=2^{-\frac{j}{2}}\psi \left(2^{-j}t-k\right),j,k\in Z$$

The signal passes through a decomposed high-pass filter and a decomposed low-pass filter. The high-frequency component of the corresponding signal is output by the high-pass filter, which is called the detail component. The output of the low-pass filter corresponds to the relatively low-frequency component of the signal, which is called the approximate component [86]. In general, the short-term volatility of stock is often affected by various information and possesses the characteristics of short-term noise. The labeling method used in the third part of the present research is based on the continuous trend characteristics of stocks to label the data and generate training datasets. Therefore, noisy data may have an obvious impact on the labeling process. It is hoped that the continuous trend characteristics of the stock are relatively stable, and the short-term noise can be filtered in the process of data labeling, especially for data with Gaussian white noise in the majority of cases; thus, we denoised the raw data by wavelet transform, and labeled the data based on the denoised data to generate training dataset. The use of DWT to denoise stock data generally requires the selection of wavelet basis, the number of decomposition layers, and the selection of threshold [79]. The DWT-based denoising process is illustrated in Figure 2. In the selection process of wavelet function, we used dozens of stocks to test the whole experimental process with different wavelet functions. The final trend prediction results are better with the wavelet function of db8. Then, we set the wavelet function as db8 with the threshold parameter of 0.04.

### 2.3. The Proposed Hybrid Method

In the current research, the main purpose is to propose a hybrid method for stock prediction, including ELM model and wavelet transform denoising. Hence, first, we imported the raw stock data, then used the labeling method to label the data, performed feature preprocessing on the raw data based on the Equations (17) and (18), and then allocated the data into training datasets, validation datasets, and test datasets. Afterwards, the training datasets were used for the training of the proposed denoised ELM (DELM). Besides, we trained the ELM model and the 12 common algorithms using the training datasets based on the raw data, and the results on the corresponding test sets were employed to compare with the results of the DELM to examine the positive influence of DWT on the ELM classification results (i.e., C1 in Figure 3). The ELM-associated parameters are shown in Table 1. Finally, the results of the 12 common algorithms were compared with those of the DELM method (i.e., C2 in Figure 3).

We used the features that were extracted by DWT-based denoising to train the DELM, and the parameters used were consistent with those applied in training of the ELM model with the raw data. Table 1 summarizes parameters required for training of the ELM model.

## 3. Feature Engineering for Stock Trend Prediction

In this section, we mainly introduce the labeling method that is used to forecast the stock trend. Through this labeling method, we can clearly define the rising and falling trend of the stock. We introduce the data set used in this paper, the statistical metrics of the experimental results and some considerations of selecting these statistical metrics.

### 3.1. Labeling Method

In the previous research, we proposed a labeling method based on the continuous trend characteristics of the time-series data [87]. In the current study, this labeling method was used to label the stock data, and then, training datasets and test datasets were generated for prediction of trends of the corresponding stocks.

In this paper, training datasets and test datasets were generated based on the closing price of transaction data. Firstly, we expanded the dimension of the closing price in order to make the current training vector of the historical stock data with the parameter length of λ. The process is formulated in Equation (15), where *x* represents the original closing price sequence, *X* denotes the matrix after dimension expansion, and each row of the *X* data represents a vector. Equation (16) indicates the labeling of these vectors, which are calculated by Equation (22).
(15)$$x=\;\left[\begin{array}{l} x_{1}\\ x_{2}\\ .\\ .\\ x_{N-1}\\ x_{N} \end{array}\right]\to X=\;\left[\begin{array}{lllll} x_{\lambda } & x_{\lambda -1} & x_{\lambda -2} & \ldots & x_{1}\\ x_{\lambda +1} & x_{\lambda } & x_{\lambda -1} & \ldots & x_{2}\\ . & . & . & \ldots & .\\ . & . & . & \ldots & .\\ x_{N-1} & x_{N-2} & x_{N-3} & \ldots & x_{N-\lambda }\\ x_{N} & x_{N-1} & x_{N-2} & \ldots & x_{N-\lambda +1} \end{array}\right]$$
(16)$$y=\left[\begin{array}{l} {label}_{\lambda }\\ {label}_{\lambda +1}\\ .\;\\ .\;\\ {label}_{N-1}\;\;\;\;\\ {label}_{N}\; \end{array}\right]$$

After expanding the dimension of the raw closing price data based on the parameter λ, we carry out the basic feature processing for the expanded data, so that the processed data features are stable, consistent with the standardization process, as summarized in Equations (17) and (18), where *x_ij_* represents a certain closing price of matrix *X*, and *M^λ^_s_* denotes the mean value of the sliding parameter λ. In this way, the data feature processing uses historical data only, and there is no look-ahead bias [88]. In the section of experiments, we attempted to examine the stationarity of the processed data. λ was set as 11, consistent with the study [87].
(17)$$f_{ij}=(x_{ij}-{M^{\lambda }}_{i})/{M^{\lambda }}_{i},x_{ij}\in X$$
(18)$${M^{\lambda }}_{s}=\frac{\sum_{i=s}^{s+\lambda -1}x_{i}}{\lambda },\;x_{i}\in x,s=1,2\ldots N-\lambda +1$$

Then, the relative maximum and minimum values of the time-series data are defined in Equation (19) with respect to the fluctuation parameter ω (the labeling parameter), and the continuous trend characteristics of the corresponding stocks are calculated according to Equations (20) and (21). Finally, the labels of the data can be obtained by Equation (22).
(19)$$h=\;\left[\begin{array}{l} h_{1}\;\\ h_{2}\\ .\;\\ .\;\\ h_{t-1}\;\;\;\;\\ h_{t}\; \end{array}\right]l=\left[\begin{array}{l} l_{1}\\ l_{2}\\ .\;\\ .\;\\ l_{m-1}\;\;\;\;\\ l_{m}\; \end{array}\right]$$
(20)$$TD(h_{i}l_{i}-1)=abs(\frac{h_{i}-l_{i}-1}{l_{i}-1}),i>1$$
(21)$$TD(l_{i}h_{i}-1)=abs(\frac{l_{i}-h_{i}-1}{h_{i}-1}),i>1$$
(22)$$x^{label}=\left\{ \begin{array}{l} 1,\;\;\;\;\;\;if\;\;\;\;\;\;\;\;x\in \;\;\left\{\begin{array}{l} x|l_{i}-1\leq x_{0}\leq h_{i},TD(h_{i}l_{i}-1)\geq \omega ,\\ i=2,3,4\ldots t;x_{0}\in \;\{x_{j}|j=0\},j=0,1,2\ldots \lambda \end{array}\right\}\\ 0,\;\;\;\;if\;\;\;\;\;\;\;\;x\in \;\;\left\{\begin{array}{l} x|h_{i}-1\leq x_{0}\leq l_{i},TD(l_{i}h_{i}-1)\geq \omega ,\\ i=2,3,4\ldots m;x_{0}\in \;\{x_{j}|j=0\},j=0,1,2\ldots \lambda \end{array}\right\} \end{array}\right.$$

### 3.2. Datasets

The stock data used in the current research are from a pool of hundreds of stocks in the Shanghai and Shenzhen stock markets in China, covering various industries. The date of trading these stocks backs to 1 January 2001 to 3 December 2020, lasting for approximately 20 years. The transaction data of each trading day are taken as the raw data, including stock code, opening price, the highest price, the lowest price, closing price, trading volume, etc. Some suspended stocks or newly listed stocks are deleted, and 400 stocks with more than 4000 rows of data are screened out as our dataset. The data are from https://tushare.pro/ (accessed on 2 January 2021), which can be downloaded in the sub-category of “Backward Answer Authority Quotes” under the category of “Quotes Data”. The data can also be downloaded for free through https://github.com/justbeat99/400_stocks_data_zips.git (accessed on 2 January 2021). After downloading the raw data, we performed feature preprocessing on the data according to Equations (17) and (18), and then labeled the data according to Equations (20)–(22) to generate labeled datasets, and segmented each stock data with the first 70% of the date for the training dataset, 15% in the middle part for the validation dataset, and the last 15% for the test dataset. As a result, the training, validation, and test datasets of 400 stocks with labeled data could be obtained. We checked the balance of the positive and negative samples on the training, validation, and test datasets of these 400 stocks, and it was found that all the datasets were relatively balanced. The balance table is submitted in the Appendix A. Appendix B
Table A2 provides the balanced datasets for some stocks. It can be seen from Appendix B
Table A2 that for the listed data, the training datasets are half of the positive and negative samples, i.e., they are all relatively balanced. Regarding the validation datasets, the proportion of positive samples for 000005 is 39%, the proportion of positive samples for 000025 is 34%, and the proportion of positive samples for 000520 is 31% that are imbalanced. Regarding the test datasets, the proportion of positive samples for 000055 is 35%, the proportion of positive samples for 000068 is 37%, and the proportion of positive samples for 000523 is 39%. The balance of these situations is slightly worse. However, they all happen in the validation dataset or test dataset of a small number of stocks, and their impact is not great. We further checked the data of all stocks, and it was found that the training dataset was basically balanced. The sample balance sheet for positive and negative cases for 400 stocks is submitted as Appendix A.

### 3.3. Statistical Metrics

Since our datasets are relatively balanced, five common statistical metrics were employed to evaluate the classification effect, including Accuracy (Acc), Recall (R), Precision (P), F1 score (F1), and area under the curve (AUC) [89], as shown in Table 2.

In Table 2, TP represents the correctly predicted proportion of positive samples; FN denotes the incorrectly predicted proportion of positive samples; FP represents the proportion of negative samples that are predicted incorrectly; and TN demonstrates the proportion of negative samples that are predicted correctly [90]. In terms of AUC, $x_{i}^{+}$ and $x_{i}^{–}$ represent data points with positive and negative labels, respectively. Besides, *f* is a general classification model, 1 is an indicator function that is equal to 1 when $f\left(x_{i}^{+}\right)\geq f\left(x_{j}^{–}\right)$; otherwise that is equal to 0; *N*^+^ (resp., *N^–^*) is the number of data points with positive (resp., negative) labels, and *M = N*^+^*N*^–^ denotes the number of matching points with opposite labels (*x*^+^, *x*^–^), with a value ranging from 0 to 1 [91].

Acc is the most basic evaluation metric, which mainly reflects the correctness of the forecasting as a whole [92]. It simply calculates the ratio, while it does not differentiate categories. Because type of error costs may be different, it is not advised to use only Acc to measure the case of unbalanced samples, because the generalizability of the model and the random prediction problem caused by sample skew are not considered. Generally speaking, the higher the Acc, the better the classifier. Our datasets are relatively balanced datasets, therefore, Acc is a promising evaluation metric. Precision is a measure of accuracy that represents the proportion of examples that are classified as positive examples but actually positive examples. Recall is a measure of coverage, which is used to measure the proportion of positive cases that are correctly classified as positive cases. F1 takes into account the Acc and Recall of the classification model [93], and can be regarded as the harmonic average of the accuracy and recall of the model. The physical meaning of AUC is the probability of taking any positive/negative case, and the positive case ranks before the negative case. The AUC reflects the sorting ability of classifiers. It is noteworthy that the AUC is not sensitive to whether the sample categories are balanced or not, justifying why performance of a classifier is typically evaluated by the AUC for unbalanced samples [94]. For a specific classifier, it is impossible for us to simultaneously improve all the above-mentioned metrics. However, if a classifier can correctly classify all instances, all metrics are optimal. Therefore, we mainly considered the actual effects (i.e., the results of Acc, the sensitivity of balanced samples, and the values of the AUC).

## 4. Experiments

In the section, we visualized and analyzed the results of the DWT. The stationarity of the feature data is tested. The labeling process of the labeling method is described in detail through the visualization. The results of ELM and DELM are compared and analyzed. Finally, the results of DELM are compared with these of some common classification algorithms.

### 4.1. The Visualization of the DWT-Based Denoising

After completing the DWT-based denoising process, we could obtain the denoised data. The raw data and the denoised data are checked. As shown in Figure 4, the line charts of raw data and denoised data of four stocks can be observed. Since the number of the raw data and the denoised data exceeds 4000, the visualization of all the data in one graph is not very intuitive. We partially enlarged the graph of the last 300 data for each stock. As displayed in Figure 4, the trend of the time-series data after denoising is smoother, and the result of the continuous trend characteristics is more stable. An abnormal fluctuation in the raw time-series data is often caused by random accidents. Abnormal fluctuations in stock market caused by accidents often reflect short-term surges and plunges, causing stock price to deviate from the normal trend. However, when accidents pass, the stock price often returns to the original normal trend. Therefore, it is hoped that the denoised data can filter such abnormal noise and maintain better trend continuity. It can be seen from the Figure 4 that the denoised data are basically less sensitive to such abnormal points, improving the continuity of the trend after denoising the data. This result is in line with our needs and expectations. It was also noticed that after denoising the data, the relatively high-price point was basically lower compared to the raw data in a local area, and the relatively low-price point was higher compared to the raw data in a local area. This is also in line with our needs for denoising. It is hoped that the DWT-based denoising can enhance the trend characteristics of stocks, and then better train machine learning models for predicting changes in the trend of stocks.

### 4.2. Testing of Stationarity

In general, it is highly essential to standardize or normalize the data before the data are used to train the model, so that the data can be mapped to a relatively stable fluctuation interval with a relatively stable volatility, which is convenient for a model to learn such a norm according to the eigenvector. The traditional standardization or normalization methods are used to process all the data [95], which are not appropriate for the time-series data and have the problem of look-ahead bias [96]. The raw data processed by Equations (17) and (18) were stable. It is necessary to conduct a stationarity test on the data processed by Equations (17) and (18) after DWT-based denoising to indicate whether there is a stable sequence, which is convenient for machine learning models to learn. Stationary data could improve the prediction ability of machine learning models. Figure 5 shows the results of feature processing based on Equations (17) and (18) for the DWT-based denoised data of stock code 000005. Figure 5a illustrates the sequence diagram of the denoised data. Figure 5a shows that the denoised data are not stable and do not have a stable fluctuation form. Figure 5b displays the featured data after the denoised data are processed by the Equations (17) and (18). It can be seen from Figure 5b that after feature processing, the data are mapped to a relatively stable fluctuation interval, and the mean is around zero. Figure 5c is the autocorrelation graph, and Figure 5d is the partial autocorrelation graph. It can be seen that once the lag parameter exceeds 15, the corresponding autocorrelation value and partial autocorrelation value fluctuate around zero. It can be concluded that, basically, the features obtained by the Equations (17) and (18) from denoised data are stable (Figure 5). In order to obtain the exact results from the statistical level, we conducted an enhanced ADF test on the 400 stocks [97,98]. Appendix B
Table A3 shows the results of ADF test for some stocks. Others are submitted as Appendix A. From Appendix B
Table A3, we can see that the statistic based on ADF test of 000005 is −10.77, which is less than the critical values of 1% (−3.43), 5% (−2.86), and 10% (−2.57). Simultaneously, the *p* value is 2.37 × 10^−19^, which is close to zero. From the results of the ADF test, it can be observed that the features processed by Equations (17) and (18) from DWT-based denoised data are stationary. All the 400 stocks were checked, and these stationarity data for all stocks are consistent, i.e., these are all stable sequences.

### 4.3. Labeling Process

The labeling method used in this paper is based on the continuous trend features of the corresponding stock. In the process of labeling the stock data, the volatility parameter ω needs to be given, based on *ω*, the relatively high- and low- price points are calculated according to the Equations (19)–(21), and the continuous trend indicator TD is calculated based on these high- and low-price points. The labeling method does not remarkably care about the short-term normal fluctuations of stock price, while it is more concerned with long-term, continuous trends. Because it uses the relatively high- and low- price points in a period to calculate the TD, the correct calculation of these price points for the corresponding period determines the labelling of all the data in such points. Then, if the stock price fluctuation caused by some short-term random events exceeds the threshold *ω*, i.e., the short-term rises and falls suddenly and sharply, which is caused by accidental events, the labeling method may regard this fluctuation as a trend of rise or fall, and then label the data and use the labeled data for training the model. The worst result is that the labelled data in this period may be biased (especially if the price returns to normal after the abnormal event), which may cause biased training results of the model.

Therefore, we used the DWT-based denoising algorithm to denoise the raw stock data in order to obtain the denoised data with better continuous trend characteristics, i.e., the denoised data can smooth such abnormal fluctuations, and then, change the relative high- and low-price points compared to the raw data for the corresponding period. Figure 6 shows the visualization process of the labeling of stock code 000005 with *ω* equal to 0.05. The red line indicates a downward trend and green line shows an upward trend. Figure 6a displays the labeling process of the DWT-based denoising data, and Figure 6b illustrates the labeling process of the raw data. Due to the denser data, it is roughly found that the continuity of the labeling process of denoised data is superior. The labeling process of the last 300 data from the two datasets was partially enlarged (Figure 6c,d). It can be clearly seen that the data that underwent DWT denoising are labeled with the labeling method based on the proposed continuous trend feature, reflecting better trend continuity. In particular, in the section circled by the orange box in the lower right corner, it can be seen that in the c plot, the labeling method is applicable to all the corresponding stock trends as a downtrend, while the two small rebounds in plot d are labeled as green, indicating that they are uptrend. The difference in the results originates from the difference in the sensitivity of the calculation of the high- and low-price points during this period. Similarly, the same situation exists for the orange box in the upper left corner. In the plot c, the labeling method labels the trends of this period as (fall, rise, fall), and in plot d, trends are indeed labeled as (fall, rise, fall, rise, fall). It can be clearly seen that the smoothness of the data after denoising is better, and the labeling process is more realistic and better reflects the characteristics of the continuous trend. In plots c and d, the serial numbers from 100 to 200 achieve the same conclusion mentioned above. The difference in labeling caused by noise may appear in training of an oversensitive model, and the accuracy of prediction and other metrics may also be sensitive to the validation set and test set. Therefore, it can be seen from the graphical results that the data after DWT-based denoising possess good smoothness and continuous trend characteristics, which can better improve the labeling effect of our labeling method. The trend continuity of the labeled data is better, reflecting the continuous trend characteristics of the corresponding stocks. This is more in line with actual investment behavior.

### 4.4. Resluts of ELM and DELM

After analyzing the smoothness of the data denoised by DWT and the stationarity of the data after feature processing by Equations (17) and (18), the obtained features were used to train the DELM. In order to better evaluate the effects of DWT on the final results, we established two models: the ELM model based on feature training of raw data, and the DELM based on feature training of denoised data. In the ELM training phase, we used the “High-Performance Extreme Learning Machines” toolbox [99]. Table 3, Table 4, Table 5, Table 6 and Table 7 present the results of Acc, P, R, F1, AUC of the two models with some stocks. The results in the validation dataset were mainly used to verify the selection of relevant parameters, and to prevent problems, such as overfitting a model trained on the training dataset. We analyzed the results in the test dataset with concentration of the results in the validation dataset. As shown in Table 3, in terms of the Acc, the results of the DELM in the test dataset significantly exceed those of the ELM model. For each stock, the Acc value of the DELM was higher than that of the ELM model for all stocks. The Acc of stock code 000007 increased from 0.5770 to 0.6483, with an increase of seven percentage points; the Acc of 000025 also increased from 0.6057 to 0.6894, with an increase of eight percentage points; the results of the improvement for codes 000048 and 000402 were not significantly different. With the Acc of 000520, the DELM significantly increased the result of the ELM model from 0.6225 to 0.7565, with an increase of 13%. In addition, the Acc for code 000530 was elevated by 10%. Other stocks’ Acc improvement was basically five to six percentage points. The above-mentioned results were then averaged. The mean values showed that the DELM increased the ELM’s Acc from 0.6445 to 0.6909, with an average increase of more than five percentage points. This is also in line with the average improvement result in the validation dataset, and the average improvement in the validation dataset is about three points.

From Table 4, it can be seen that in terms of the value of the Precision metric, the values are not as improved as the Acc metric. The results of the Precision metric for the ELM model are partly good, and some represent the results of the DELM, e.g., the stock codes of 000005, 000150, 000151, 000402, 000404, 000420, 000430, 000507, and 000509. However, in general, it can be seen from the average results that the values of the Precision metric for the DELM increased from 0.6170 to 0.6506, indicating improvement to a certain degree.

Regarding the values of the Recall metric, it can be seen from Table 5 that it is not significantly improved, and even in a variety of cases, the values of the Recall metric for the ELM model are higher. From the mean values of Recall metric, it can be seen that the mean values of Recall metric for the ELM model dropped by about 0.53% compared to the DELM.

Regarding the values of F1 metric presented in Table 6, we can also achieve the same conclusion as Recall metric. For each stock, the two models possess their own results. For the mean value, it was elevated by about 1.2 percentage points.

The values of the AUC metric are presented in Table 7. As far as the AUC values of the ELM and DELM were concerned, the AUC value was not improved in only one sample for the DELM, i.e., the AUC value of 0.6387 for the ELM model for code 000005 was higher than the AUC value for the DELM for code 0.6108. From the mean value of the AUC, it can be seen that the AUC value for the ELM model compared with the DELM was elevated from 0.6447 to 0.6856, with an increase of four percentage points, which is very significant.

At the same time, we calculated the mean values of the statistical metrics for all 400 stocks presented in Table 8 (the values for other stocks are submitted as Appendix A). From Table 8, it can be seen that the conclusion is basically the same. The values of Acc and AUC for the DELM have been significantly improved compared to the ELM model. The improvement of F1 for the DELM is not statistically significant (0.6343 versus 0.6369). It was also noticed that the P value of the ELM model improved (within 3.7%) compared with that of the DELM. The value of R metric decreased by an average of 2.4 percentage points. The average value of AUC rose from 0.6517 to 0.6892, with an increase of 3.75 percentage points. In the section of statistical metrics, we compared the differences between the different metrics, and concentrated on the values of Acc and AUC. The results of the 400 stocks were also checked, as presented in the Appendix A, and it was found that in terms of the values of Acc and AUC metrics, for the DELM, the values for each stock were elevated, indicating that DWT-based denoising could remarkably improve the ELM model prediction results based on the labeling method. It was demonstrated that the interpretation regarding the combination of continuous trend-based labeling method and DWT-based denoising was correct. The results showed that the denoised data were highly appropriate for the continuous trend-based labeling method. The results also highlighted the rationality and superiority of the architecture of the proposed hybrid method.

### 4.5. The DELM Method and Other Classification Algorithms

In order to further verify the prediction ability of the proposed hybrid method (DELM), we also tested prediction ability of other common models in datasets of the 400 stocks. Among them, the training of deep learning models (e.g., recurrent neural network (RNN) and LSTM), with good dealing with time-series data problems, was carried out through Pytorch (ver. 1.5.1) [100], and the other models were trained using the Sklearn toolkit (ver. 0.23.2) [101]. All the models and associated parameters are summarized in Table 9. The parameters’ names and specific functions are not detailed here (please refer to the related literatures).

Due to space limitations, we presented the results of only six stocks, and the results of other stocks are submitted as Appendix A. As shown in Table 10, among the results of all six stocks, the Acc values of the DELM were most promising, which significantly surpassed the Acc values of other common models, basically reaching an accuracy rate of 0.7. Additionally, the Acc values of the DELM surpassed the results of other common models, and again verified the efficacy of DWT-based denoising. In addition, the results of LSTM, RNN, RF, GBT, ABT, and SVC were relatively better than those of other common models (basically between 0.67 and 0.68). The values of the AUC metric were concerned, those of stock codes 000005, 000007, and 000025 for the DELM were not the best in all algorithms, and the most reliable were found in other stock codes. As far as the Acc values were concerned, these values in the proposed DELM were the best on all the stock codes, and the AUC values of the proposed DELM were the best among all the other algorithms. In addition to the six stocks, we checked the results of all 400 stocks, and the conclusions were basically consistent with those of the six stocks, i.e., the proposed DELM could significantly improve the values of Acc and AUC in prediction process. Regarding the values of P, R, and F1, as explained in the section of statistical metrics, in general, each model possesses its unique advantages. This paper mainly concentrated on the values of Acc and AUC. Therefore, it is concluded that the proposed hybrid method DELM can better predict changes in the continuous trend of stocks and has a higher prediction accuracy and AUC value.

## 5. Conclusions

This research proposed a hybrid method for the trend prediction of stocks based on ELM and wavelet transform denoising. The raw data were first denoised based on DWT, feature preprocessing was performed after denoising data, and then, the DELM was trained based on the denoised data with the obtained features. Finally, the training DELM was compared with the initial ELM model on a dataset of 400 stocks. The prediction results greatly improved the values of the Acc and AUC metrics. The results fully proved the superiority of the DELM, and also showed that wavelet transform could improve the prediction ability of the ELM model. At the same time, the logical relationship between DWT-based denoising and the labeling method was analyzed based on the continuous trend, and logical explanations were provided for good results. Additionally, in order to better assess the efficacy of the proposed DELM, the predictive results of the DELM for the stocks were also compared with those of the 12 common algorithms, which the proposed DELM method outperformed. The results confirmed the superiority of the proposed DELM method as well. However, this paper does not investigate the influence of different wavelet function denoising results on improving the accuracy of stock trend prediction in-depth, which remains to be investigated by future research work.

## Figures and Tables

**Figure 1 entropy-23-00440-f001:**
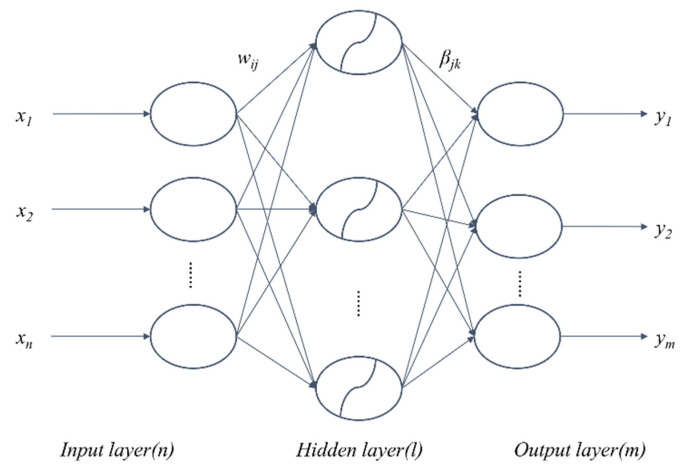
Schematical presentation of a single-hidden-layer feedforward neural network (SLFN).

**Figure 2 entropy-23-00440-f002:**
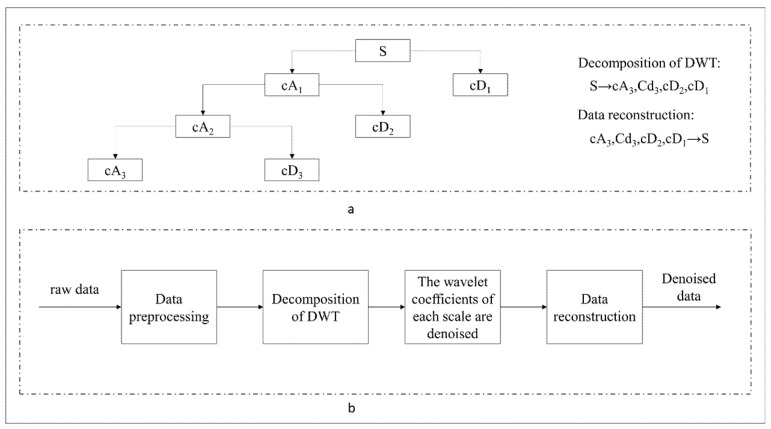
Discrete wavelet transform (DWT)-based denoising process. cA denotes the approximate component, and cD represents the detail component. Plot (**a**) shows the process of signal decomposition and reconstruction. Plot (**b**) shows the whole denoising process.

**Figure 3 entropy-23-00440-f003:**
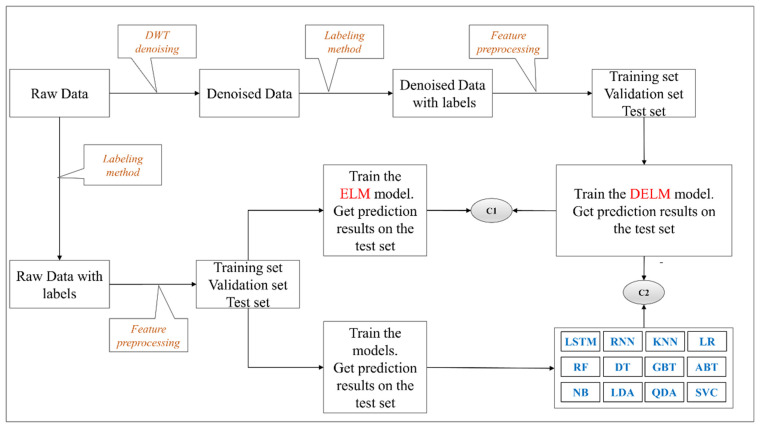
Flowchart of the DELM method.

**Figure 4 entropy-23-00440-f004:**
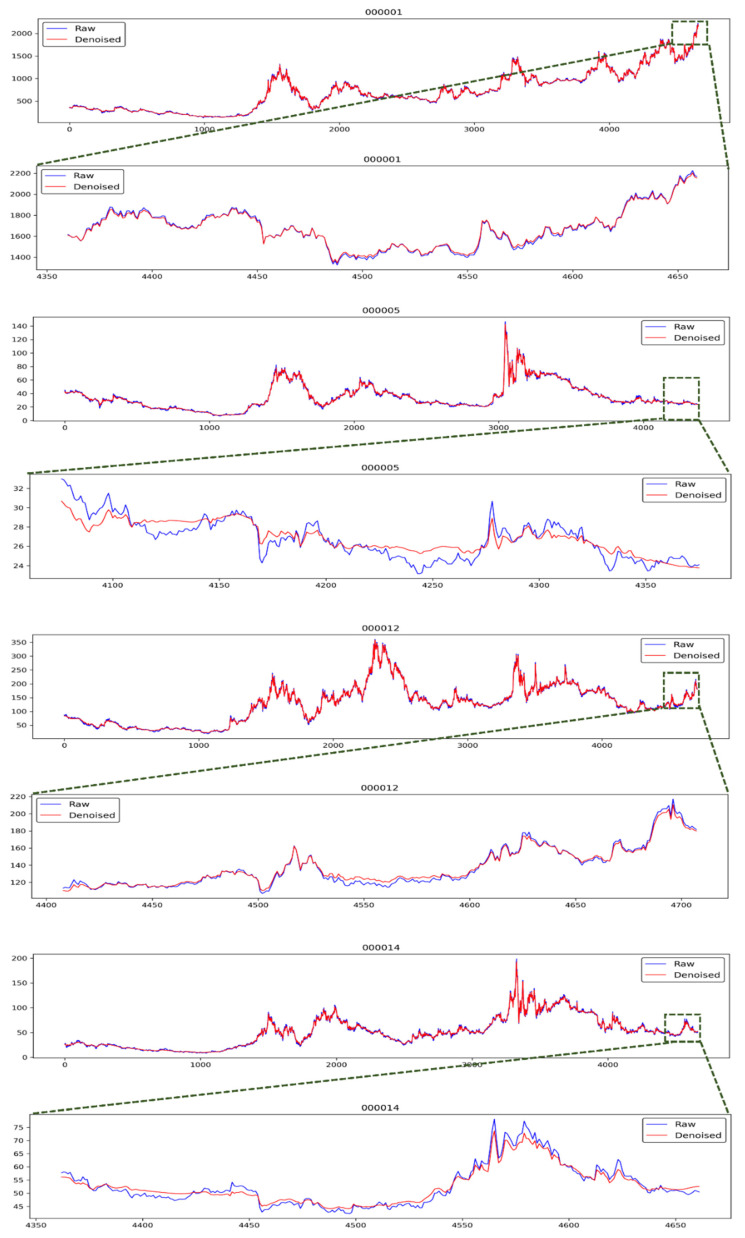
The diagrams of the four stocks using raw data and denoised data by DWT.

**Figure 5 entropy-23-00440-f005:**
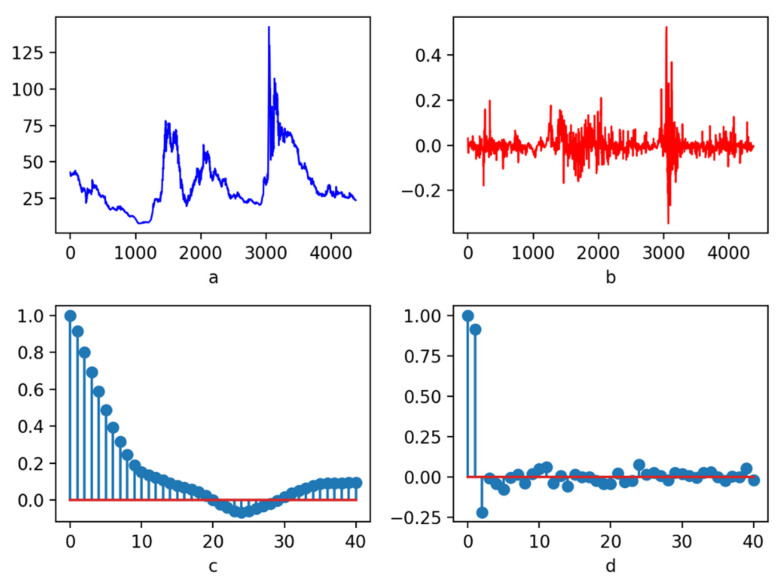
Stock data feature diagram of code 000005. Where, (**a**) is the sequence diagram of the denoised data, and the *Y* axis represents the closing price after denoising; (**b**) displays the feature diagram of time-series data after the denoised data could be processed by Equations (17) and (18), and the *Y* axis represents the value of the feature; (**c**) shows the autocorrelation graph, and the *Y* axis represents the autocorrelation value; (**d**) illustrates the partial autocorrelation graph, and the *Y* axis represents the partial autocorrelation value. The *X* axis in (**a**,**b**) represents the date, and in (**c**,**d**) represents the lag parameter.

**Figure 6 entropy-23-00440-f006:**
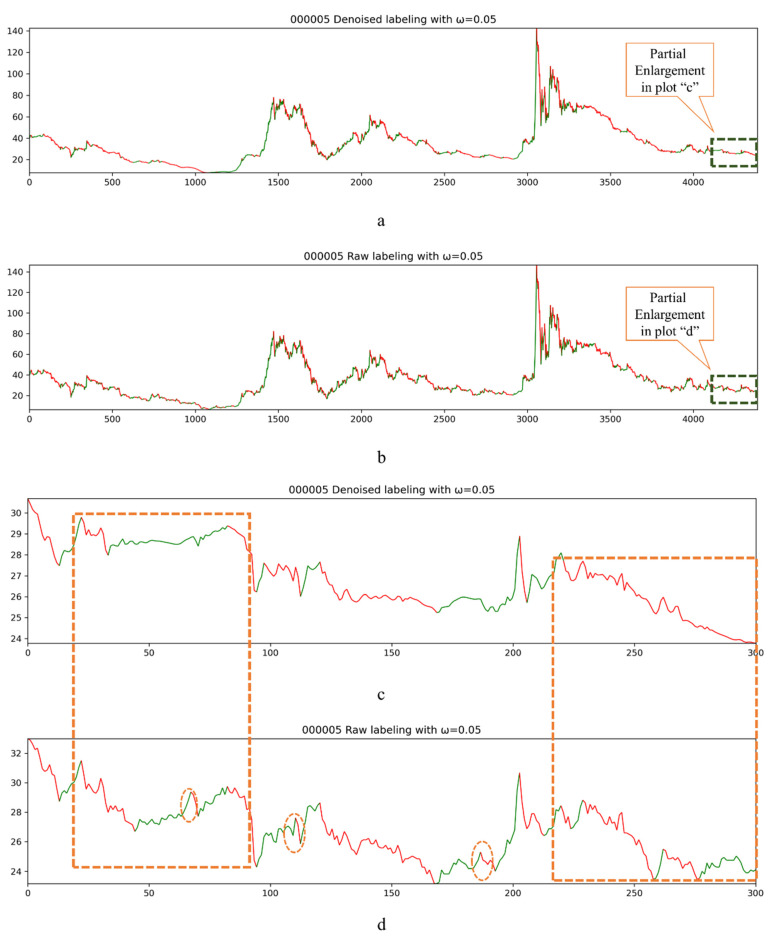
The labeling process. The *Y*-axis represents the closing price of the corresponding stock, and the *X*-axis indicates the date. The green line indicates that the labeling method is significant in labeling this segment of data as an upward trend, and the red line indicates that the labeling method can be used to label this segment of data as a downward trend. (**a**) displays the labeling process of the DWT-based denoising data, and (**b**) illustrates the labeling process of the raw data. The labeling process of the last 300 data from the two datasets was partially enlarged (**c**,**d**).

**Table 1 entropy-23-00440-t001:** Parameters required for training of the extreme learning machine (ELM) model.

Name	Input Neurons	Hidden Neurons	Activation Function	Hidden Layers	Output Neurons
First hidden layer	11	50	Sigmoid	1	50
Second hidden layer	50	50	RBF	1	1

**Table 2 entropy-23-00440-t002:** Metrics used for evaluation of classification effects.

Metrics	Formula	Evaluation Focus
Accuracy (Acc)	$\frac{TP+TN}{TP+FN+FP+FN}$	The ratio of correctly classified samples to total samples.
Recall (R)	$\frac{TP}{TP+FN}$	Proportion of correctly classified results among the true positive samples.
Precision (P)	$\frac{TP}{TP+FP}$	Proportion of correctly classified results among the results predicted to be positive samples.
F1_score (F1)	$2\times \frac{Precision\times Recall}{Precision+Recall}$	The harmonic average of precision and recall, and its value is closer to the smaller value of Precision and Recall.
AUC	$\frac{\sum_{i=1}^{N+}\sum_{j=1}^{N-}1_{f(x_{i}^{+})\geq f(x_{j}^{-})}}{M}$	The area under the Roc curve is between 0.1 and 1. Area under the curve (AUC) as a value can intuitively evaluate the quality of the classifier. The larger the value, the better the results will be.

**Table 3 entropy-23-00440-t003:** The Acc metric of corresponding validation datasets and test datasets of ELM and DELM for some stocks.

Code	ELM		DELM	
	Validation Acc	Test Acc	Validation Acc	Test Acc
000001	0.5794	0.6638	0.7096	0.6953
000005	0.6311	0.6499	0.7241	0.6941
000007	0.6062	0.5770	0.6726	0.6483
000012	0.6275	0.6506	0.6728	0.6973
000014	0.6452	0.6486	0.6166	0.7000
000025	0.5156	0.6057	0.6662	0.6894
000026	0.6370	0.6276	0.6582	0.6996
000031	0.6568	0.6716	0.7128	0.7231
000032	0.6098	0.6400	0.6261	0.7037
000048	0.6224	0.6822	0.6254	0.6837
000050	0.5912	0.6197	0.6309	0.6814
000055	0.6433	0.6124	0.6713	0.6699
000056	0.6141	0.6318	0.6686	0.7025
000061	0.6088	0.6441	0.6618	0.6897
000065	0.6494	0.6320	0.6349	0.6912
000068	0.6502	0.6095	0.6836	0.6587
000090	0.5733	0.6619	0.6117	0.6903
000150	0.6433	0.6692	0.6799	0.7317
000151	0.6336	0.6508	0.6545	0.6732
000155	0.6512	0.6352	0.6608	0.6976
000158	0.6357	0.6519	0.6143	0.6790
000402	0.6306	0.6826	0.6306	0.6896
000404	0.6432	0.6700	0.6953	0.6953
000411	0.6040	0.6346	0.6437	0.7034
000420	0.6516	0.6238	0.6275	0.6436
000422	0.6700	0.6743	0.7226	0.6885
000430	0.6235	0.6882	0.6088	0.6941
000507	0.6172	0.6398	0.6554	0.6822
000509	0.6401	0.6369	0.6260	0.6870
000519	0.6417	0.6584	0.6681	0.6935
000520	0.6716	0.6225	0.7369	0.7565
000523	0.6508	0.6494	0.7215	0.7071
000524	0.6148	0.6555	0.6294	0.6846
000526	0.6288	0.6319	0.6209	0.6367
000530	0.6445	0.6436	0.6969	0.7440
000531	0.6614	0.6489	0.6369	0.6964
000532	0.6295	0.6524	0.6552	0.6609
Mean	0.6283	0.6445	0.6603	0.6909

**Table 4 entropy-23-00440-t004:** The precision metric for the corresponding validation datasets and test datasets of ELM and DELM for some stocks.

Code	ELM		DELM	
	Validation P	Test P	Validation P	Test P
000001	0.7547	0.7074	0.6987	0.7425
000005	0.5371	0.6524	0.4686	0.5909
000007	0.5558	0.5108	0.5955	0.6316
000012	0.7035	0.6523	0.7000	0.7380
000014	0.6743	0.6462	0.6481	0.6708
000025	0.3599	0.5072	0.4686	0.5693
000026	0.6879	0.5975	0.6781	0.6205
000031	0.6119	0.6678	0.6381	0.7120
000032	0.5926	0.6503	0.6355	0.7486
000048	0.6172	0.6629	0.6139	0.7273
000050	0.6437	0.6828	0.6638	0.7437
000055	0.6090	0.4638	0.6400	0.5248
000056	0.6250	0.5776	0.6317	0.6947
000061	0.6337	0.6273	0.5817	0.6061
000065	0.6796	0.5711	0.6402	0.7300
000068	0.5878	0.4749	0.6398	0.4868
000090	0.6076	0.6906	0.6349	0.7356
000150	0.5994	0.5052	0.6457	0.5000
000151	0.5895	0.6573	0.6250	0.5749
000155	0.6793	0.5710	0.6641	0.6045
000158	0.6384	0.6260	0.6239	0.7045
000402	0.6732	0.6952	0.6690	0.6580
000404	0.7310	0.6655	0.7546	0.5248
000411	0.5662	0.6111	0.6266	0.6332
000420	0.6569	0.5195	0.5862	0.4848
000422	0.6749	0.7236	0.7683	0.8235
000430	0.6319	0.6690	0.5429	0.6289
000507	0.6650	0.6411	0.7046	0.6062
000509	0.6160	0.5385	0.6098	0.5020
000519	0.6128	0.6825	0.6498	0.7782
000520	0.4700	0.6360	0.4350	0.7057
000523	0.6192	0.5319	0.7094	0.5525
000524	0.5849	0.6364	0.6166	0.7429
000526	0.6530	0.6608	0.6779	0.7254
000530	0.6493	0.6129	0.7642	0.7021
000531	0.6218	0.6456	0.5867	0.6810
000532	0.6056	0.6580	0.6376	0.6676
Mean	0.6222	0.6170	0.6345	0.6506

**Table 5 entropy-23-00440-t005:** The values of the Recall metric for the corresponding validation datasets and test datasets of the ELM and DELM for some stocks.

Code	ELM		DELM	
	Validation R	Test R	Validation R	Test R
000001	0.3980	0.5504	0.5604	0.6201
000005	0.4749	0.5050	0.4824	0.3467
000007	0.7924	0.7852	0.7050	0.6755
000012	0.6005	0.5138	0.6485	0.5831
000014	0.5791	0.5217	0.5029	0.5514
000025	0.5481	0.7394	0.5091	0.4656
000026	0.7313	0.7055	0.7132	0.6573
000031	0.6656	0.6118	0.7128	0.6897
000032	0.7101	0.7500	0.6018	0.7048
000048	0.5667	0.5757	0.5706	0.5994
000050	0.5926	0.5425	0.6403	0.5852
000055	0.7126	0.6948	0.6747	0.5944
000056	0.5621	0.5461	0.6461	0.5414
000061	0.5103	0.5466	0.6357	0.6569
000065	0.7161	0.7235	0.6420	0.5356
000068	0.6097	0.5279	0.6140	0.5311
000090	0.5217	0.5581	0.5437	0.5630
000150	0.6633	0.6682	0.6544	0.6364
000151	0.7329	0.6890	0.7205	0.7932
000155	0.7472	0.7348	0.7581	0.8137
000158	0.6995	0.7067	0.5812	0.5959
000402	0.6782	0.6722	0.7066	0.7825
000404	0.6359	0.5762	0.6993	0.6883
000411	0.8156	0.8112	0.7485	0.7979
000420	0.7929	0.7986	0.7930	0.8703
000422	0.7326	0.7076	0.6931	0.5255
000430	0.7378	0.8073	0.7550	0.8286
000507	0.6667	0.6023	0.7046	0.6701
000509	0.6913	0.6314	0.6431	0.6318
000519	0.6321	0.6731	0.5825	0.6129
000520	0.4974	0.4949	0.6444	0.8430
000523	0.7703	0.7491	0.6942	0.6174
000524	0.7359	0.7304	0.6073	0.5417
000526	0.5521	0.5772	0.5403	0.5895
000530	0.7035	0.6353	0.7057	0.6856
000531	0.6894	0.6305	0.6506	0.7033
000532	0.6959	0.6959	0.6986	0.6657
Mean	0.6530	0.6484	0.6482	0.6431

**Table 6 entropy-23-00440-t006:** The F1 score metric for the corresponding validation datasets and test datasets of the ELM and DELM for some stocks.

Code	ELM		DELM	
	Validation F1	Test F1	Validation F1	Test F1
000001	0.5212	0.6191	0.6220	0.6758
000005	0.5041	0.5693	0.4754	0.4370
000007	0.6534	0.6190	0.6456	0.6528
000012	0.6479	0.5749	0.6733	0.6515
000014	0.6231	0.5773	0.5663	0.6053
000025	0.4345	0.6017	0.4880	0.5122
000026	0.7089	0.6471	0.6952	0.6384
000031	0.6376	0.6386	0.6734	0.7006
000032	0.6460	0.6966	0.6182	0.7260
000048	0.5908	0.6162	0.5914	0.6572
000050	0.6171	0.6046	0.6519	0.6550
000055	0.6568	0.5563	0.6569	0.5574
000056	0.5919	0.5614	0.6388	0.6085
000061	0.5654	0.5842	0.6075	0.6305
000065	0.6974	0.6383	0.6411	0.6179
000068	0.5985	0.5000	0.6266	0.5080
000090	0.5614	0.6174	0.5857	0.6379
000150	0.6297	0.5753	0.6500	0.5600
000151	0.6534	0.6728	0.6693	0.6667
000155	0.7116	0.6426	0.7080	0.6937
000158	0.6675	0.6639	0.6018	0.6457
000402	0.6757	0.6835	0.6873	0.7148
000404	0.6802	0.6176	0.7259	0.5955
000411	0.6684	0.6971	0.6821	0.7061
000420	0.7185	0.6295	0.6741	0.6228
000422	0.7026	0.7155	0.7288	0.6416
000430	0.6808	0.7316	0.6316	0.7151
000507	0.6658	0.6211	0.7046	0.6365
000509	0.6515	0.5812	0.6260	0.5595
000519	0.6223	0.6777	0.6143	0.6857
000520	0.4833	0.5566	0.5194	0.7683
000523	0.6865	0.6221	0.7017	0.5832
000524	0.6518	0.6802	0.6119	0.6265
000526	0.5983	0.6161	0.6013	0.6505
000530	0.6753	0.6239	0.7338	0.6937
000531	0.6539	0.6380	0.6170	0.6920
000532	0.6476	0.6764	0.6667	0.6667
Mean	0.6319	0.6255	0.6382	0.6377

**Table 7 entropy-23-00440-t007:** The AUC values in the corresponding validation datasets and test datasets of ELM and DELM for some stocks.

Code	ELM		DELM	
	Validation AUC	Test AUC	Validation AUC	Test AUC
000001	0.6115	0.6630	0.6904	0.6972
000005	0.6040	0.6387	0.6455	0.6108
000007	0.6172	0.6001	0.6769	0.6489
000012	0.6319	0.6404	0.6738	0.6940
000014	0.6461	0.6392	0.6161	0.6789
000025	0.5235	0.6274	0.6233	0.6378
000026	0.6121	0.6301	0.6525	0.6927
000031	0.6576	0.6686	0.7128	0.7212
000032	0.6095	0.6275	0.6263	0.7036
000048	0.6204	0.6713	0.6228	0.6847
000050	0.5910	0.6257	0.6301	0.6847
000055	0.6461	0.6314	0.6716	0.6525
000056	0.6139	0.6215	0.6667	0.6820
000061	0.6085	0.6365	0.6579	0.6844
000065	0.6395	0.6405	0.6348	0.6813
000068	0.6451	0.5927	0.6753	0.6266
000090	0.5758	0.6596	0.6123	0.6865
000150	0.6449	0.6690	0.6777	0.7015
000151	0.6390	0.6492	0.6564	0.6911
000155	0.6340	0.6448	0.6518	0.7135
000158	0.6327	0.6534	0.6144	0.6775
000402	0.6232	0.6828	0.6173	0.6901
000404	0.6449	0.6634	0.6946	0.6935
000411	0.6084	0.6278	0.6414	0.7125
000420	0.6319	0.6528	0.6320	0.6990
000422	0.6657	0.6680	0.7250	0.6991
000430	0.6125	0.6816	0.6235	0.7033
000507	0.6089	0.6391	0.6455	0.6804
000509	0.6414	0.6360	0.6264	0.6721
000519	0.6411	0.6573	0.6609	0.7016
000520	0.6234	0.6173	0.7038	0.7601
000523	0.6517	0.6680	0.7200	0.6845
000524	0.6172	0.6553	0.6286	0.6813
000526	0.6289	0.6332	0.6259	0.6448
000530	0.6413	0.6430	0.6949	0.7362
000531	0.6633	0.6486	0.6381	0.6966
000532	0.6309	0.6503	0.6558	0.6609
Mean	0.6254	0.6447	0.6547	0.6856

**Table 8 entropy-23-00440-t008:** The mean values of statistical metrics in validation datasets and test datasets for the ELM and DELM for the 400 stocks.

Metric	ELM		DELM	
	Validation	Test	Validation	Test
Acc	0.6386	0.6523	0.6634	0.7013
p	0.6539	0.6312	0.6811	0.6681
R	0.6648	0.6497	0.6436	0.6257
F1	0.6548	0.6343	0.6567	0.6369
AUC	0.6357	0.6517	0.6602	0.6892

**Table 9 entropy-23-00440-t009:** Related parameters for training of the 12 common models.

Models	Related Parameters
LSTM	Input size = 11; hidden size = 11; output size = 2; layer num = 1; Activation function = Relu; Optimization function = Adam with learning rate = 0.009, betas = (0.9, 0.999), eps = 1 × 10^−8^; loss function = Cross Entropy Loss; stop training epoch = 200
RNN	Input size = 11; hidden size = 11; output size = 2; layer num = 1; Activation function = Relu; Optimization function = Adam with learning rate = 0.009, betas = (0.9, 0.999), eps = 1 × 10^−8^; loss function = Cross Entropy Loss; stop training epoch = 200
KNN	n of neighbors = 5
LR	penalty = ‘l2’
RF	n of estimators = 50
DT	max of depth = 3
GBT	Learning rate = 0.1, n_estimators = 100
ABT	n of estimators = 50
NB	priors = None; var smoothing = 1 × 10^−^^8^
LDA	solver = ‘svd’; store covariance = False; tol = 1 × 10^−^^4^
QDA	store covariance = False; tol = 1 × 10^−^^4^
SVC	kernel = ‘rbf’; C = 2

**Table 10 entropy-23-00440-t010:** Classification results of the DELM and other classification algorithms on several test datasets and validation datasets.

Code	Model	Validation Acc	Test Acc	Validation P	Test P	Validation R	Test R	Validation F1	Test F1	Validation AUC	Test AUC
000001	ELM	0.5794	0.6638	0.7547	0.7074	0.3980	0.5504	0.5212	0.6191	0.6115	0.6630
	DELM	0.7096	0.6953	0.6987	0.7425	0.5604	0.6201	0.6220	0.6758	0.6904	0.6972
	LSTM	0.5711	0.6522	0.6875	0.6512	0.4662	0.6450	0.5555	0.6480	0.5897	0.6522
	RNN	0.5732	0.6694	0.7200	0.6870	0.4348	0.6222	0.5272	0.6501	0.5977	0.6690
	KNN	0.5594	0.6409	0.7217	0.6548	0.3806	0.5850	0.4984	0.6180	0.5910	0.6405
	LR	0.4893	0.6209	0.7320	0.7412	0.1766	0.3631	0.2846	0.4874	0.5445	0.6191
	RF	0.6052	0.6481	0.7716	0.6624	0.4453	0.5937	0.5647	0.6261	0.6334	0.6477
	DT	0.6223	0.6552	0.7674	0.6779	0.4925	0.5821	0.6000	0.6264	0.6453	0.6547
	GBT	0.6223	0.6810	0.7828	0.7000	0.4751	0.6254	0.5913	0.6606	0.6483	0.6806
	ABT	0.6180	0.6853	0.7668	0.7042	0.4826	0.6311	0.5924	0.6657	0.6420	0.6849
	NB	0.4864	0.6109	0.6693	0.6697	0.2114	0.4265	0.3214	0.5211	0.5350	0.6096
	LDA	0.5508	0.6423	0.7785	0.7137	0.3060	0.4669	0.4393	0.5645	0.5941	0.6411
	QDA	0.5894	0.6052	0.6471	0.5894	0.6294	0.6744	0.6381	0.6290	0.5824	0.6056
	SVC	0.6338	0.6838	0.8202	0.7218	0.4652	0.5908	0.5937	0.6498	0.6636	0.6832
000005	ELM	0.6311	0.6499	0.5371	0.6524	0.4749	0.5050	0.5041	0.5693	0.6040	0.6387
	DELM	0.7241	0.6941	0.4686	0.5909	0.4824	0.3467	0.4754	0.4370	0.6455	0.6108
	LSTM	0.5726	0.5976	0.4549	0.5684	0.4000	0.5136	0.4249	0.5385	0.5426	0.5911
	RNN	0.6405	0.6507	0.5643	0.6468	0.3965	0.5246	0.4643	0.5775	0.5981	0.6409
	KNN	0.6250	0.6134	0.5267	0.5848	0.4942	0.5382	0.5100	0.5606	0.6023	0.6076
	LR	0.6311	0.6362	0.5497	0.6535	0.3629	0.4385	0.4372	0.5249	0.5845	0.6210
	RF	0.6570	0.6499	0.5659	0.6371	0.5637	0.5482	0.5648	0.5893	0.6408	0.6421
	DT	0.5945	0.6149	0.4880	0.5839	0.5483	0.5548	0.5164	0.5690	0.5865	0.6103
	GBT	0.6204	0.6575	0.5205	0.6450	0.4903	0.5615	0.5050	0.6004	0.5978	0.6501
	ABT	0.6143	0.6423	0.5103	0.6162	0.5753	0.5814	0.5408	0.5983	0.6075	0.6376
	NB	0.5793	0.5616	0.4388	0.5631	0.2355	0.1927	0.3065	0.2871	0.5195	0.5331
	LDA	0.6311	0.6606	0.5365	0.6653	0.4826	0.5216	0.5081	0.5847	0.6053	0.6498
	QDA	0.5930	0.5951	0.4778	0.5989	0.3320	0.3522	0.3918	0.4435	0.5476	0.5764
	SVC	0.6463	0.6530	0.5534	0.6420	0.5405	0.5482	0.5469	0.5914	0.6280	0.6449
000007	ELM	0.6062	0.5770	0.5558	0.5108	0.7924	0.7852	0.6534	0.6190	0.6172	0.6001
	DELM	0.6726	0.6483	0.5955	0.6316	0.7050	0.6755	0.6456	0.6528	0.6769	0.6489
	LSTM	0.6062	0.5874	0.5540	0.5185	0.8201	0.8033	0.6611	0.6301	0.6189	0.6113
	RNN	0.5951	0.6049	0.5488	0.5332	0.7827	0.8041	0.6444	0.6404	0.6063	0.6270
	KNN	0.6045	0.5754	0.5587	0.5102	0.7405	0.7407	0.6369	0.6042	0.6126	0.5937
	LR	0.5397	0.5348	0.5049	0.4833	0.8927	0.9111	0.6450	0.6316	0.5607	0.5766
	RF	0.6256	0.6207	0.5797	0.5526	0.7301	0.7000	0.6462	0.6176	0.6318	0.6295
	DT	0.6402	0.6677	0.5965	0.6012	0.7163	0.7148	0.6509	0.6531	0.6447	0.6730
	GBT	0.6159	0.6548	0.5703	0.5785	0.7301	0.7778	0.6404	0.6635	0.6227	0.6684
	ABT	0.6207	0.6532	0.5749	0.5765	0.7301	0.7815	0.6433	0.6635	0.6272	0.6674
	NB	0.5429	0.5381	0.5073	0.4841	0.8374	0.8444	0.6319	0.6154	0.5605	0.5721
	LDA	0.5802	0.5900	0.5349	0.5197	0.7958	0.8296	0.6398	0.6391	0.5930	0.6165
	QDA	0.5997	0.5624	0.5714	0.5000	0.5813	0.5481	0.5763	0.5230	0.5986	0.5608
000012	ELM	0.6275	0.6506	0.7035	0.6523	0.6005	0.5138	0.6479	0.5749	0.6319	0.6404
	DELM	0.6728	0.6973	0.7000	0.7380	0.6485	0.5831	0.6733	0.6515	0.6738	0.6940
	LSTM	0.5271	0.6758	0.5820	0.6313	0.6077	0.7102	0.5941	0.6680	0.5137	0.6784
	RNN	0.5666	0.6898	0.6473	0.6833	0.5290	0.6065	0.5822	0.6424	0.5728	0.6836
	KNN	0.5921	0.6492	0.6780	0.6351	0.5434	0.5569	0.6033	0.5934	0.6001	0.6423
	LR	0.6686	0.6818	0.7354	0.6534	0.6551	0.6554	0.6929	0.6544	0.6708	0.6798
	RF	0.6161	0.6846	0.6844	0.6735	0.6079	0.6092	0.6439	0.6397	0.6175	0.6790
	DT	0.5935	0.6733	0.6747	0.6643	0.5558	0.5846	0.6095	0.6219	0.5997	0.6667
	GBT	0.6289	0.6931	0.7080	0.6915	0.5955	0.6000	0.6469	0.6425	0.6344	0.6861
	ABT	0.6048	0.6846	0.6987	0.6875	0.5409	0.5754	0.6098	0.6265	0.6154	0.6764
	NB	0.4986	0.6181	0.6070	0.6821	0.3449	0.3169	0.4399	0.4328	0.5239	0.5956
	LDA	0.6586	0.6535	0.7213	0.6227	0.6551	0.6246	0.6866	0.6237	0.6592	0.6513
	QDA	0.6091	0.5601	0.6584	0.5202	0.6551	0.5538	0.6567	0.5365	0.6015	0.5596
	SVC	0.6275	0.6945	0.6966	0.7011	0.6154	0.5846	0.6535	0.6376	0.6295	0.6863
000014	ELM	0.6452	0.6486	0.6743	0.6462	0.5791	0.5217	0.6231	0.5773	0.6461	0.6392
	DELM	0.6166	0.7000	0.6481	0.6708	0.5029	0.5514	0.5663	0.6053	0.6161	0.6789
	LSTM	0.6278	0.6541	0.6646	0.6530	0.5404	0.5416	0.5947	0.5883	0.6289	0.6458
	RNN	0.6020	0.5893	0.6294	0.5614	0.5203	0.5053	0.5690	0.5310	0.6031	0.5831
	KNN	0.5851	0.5729	0.6119	0.5367	0.4944	0.5217	0.5469	0.5291	0.5863	0.5691
	LR	0.6295	0.6471	0.6610	0.6364	0.5508	0.5435	0.6009	0.5863	0.6305	0.6395
	RF	0.6237	0.6229	0.6619	0.5980	0.5254	0.5497	0.5858	0.5728	0.6250	0.6174
	DT	0.6295	0.6357	0.7039	0.6314	0.4633	0.5000	0.5588	0.5581	0.6316	0.6257
	GBT	0.6295	0.6629	0.6480	0.6387	0.5876	0.6149	0.6163	0.6266	0.6300	0.6593
	ABT	0.6409	0.6329	0.6604	0.6080	0.5989	0.5683	0.6281	0.5875	0.6415	0.6281
	NB	0.5451	0.6000	0.5732	0.5946	0.3983	0.4099	0.4700	0.4853	0.5470	0.5859
	LDA	0.6223	0.6600	0.6424	0.6448	0.5734	0.5807	0.6060	0.6111	0.6230	0.6541
	QDA	0.5594	0.5729	0.5991	0.5466	0.3927	0.4193	0.4744	0.4745	0.5615	0.5615
	SVC	0.6423	0.6600	0.6733	0.6533	0.5706	0.5559	0.6177	0.6007	0.6433	0.6523
000025	ELM	0.5156	0.6057	0.3599	0.5072	0.5481	0.7394	0.4345	0.6017	0.5235	0.6274
	DELM	0.6662	0.6894	0.4686	0.5693	0.5091	0.4656	0.4880	0.5122	0.6233	0.6378
	LSTM	0.5709	0.6353	0.4006	0.5470	0.5285	0.5528	0.4553	0.5491	0.5606	0.6219
	RNN	0.6075	0.5891	0.4414	0.4880	0.5121	0.5475	0.4693	0.5093	0.5843	0.5823
	KNN	0.5881	0.5943	0.4141	0.4966	0.5146	0.5106	0.4590	0.5035	0.5702	0.5807
	LR	0.4602	0.5560	0.3593	0.4725	0.7531	0.8768	0.4865	0.6141	0.5314	0.6082
	RF	0.6023	0.6539	0.4183	0.5654	0.4393	0.6092	0.4286	0.5864	0.5627	0.6466
	DT	0.6051	0.6567	0.4325	0.5724	0.5230	0.5845	0.4735	0.5784	0.5852	0.6450
	GBT	0.6051	0.6667	0.4330	0.5714	0.5272	0.6901	0.4755	0.6252	0.5862	0.6705
	ABT	0.5881	0.6582	0.4118	0.5683	0.4979	0.6303	0.4508	0.5977	0.5662	0.6536
	NB	0.3935	0.4879	0.2793	0.4291	0.4979	0.8204	0.3579	0.5635	0.4188	0.5420
	LDA	0.5270	0.5759	0.3917	0.4830	0.7113	0.7500	0.5052	0.5876	0.5718	0.6042
	QDA	0.6435	0.5787	0.4737	0.4586	0.4519	0.2535	0.4625	0.3265	0.5969	0.5258
	SVC	0.5710	0.6667	0.4060	0.5671	0.5690	0.7289	0.4739	0.6379	0.5705	0.6768

## Data Availability

The data are from https://tushare.pro/ (accessed on 2 January 2021), which can be downloaded in the sub-category of “Backward Answer Authority Quotes” under the category of “Quotes Data”. The data can also be downloaded for free through https://github.com/justbeat99/400_stocks_data_zips.git (accessed on 2 January 2021).

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
