# Peer review of "A Hybrid Method Based on Extreme Learning Machine and Wavelet Transform Denoising for Stock Prediction"

_entropy, 2021, doi:10.3390/e23040440_

Round 1
Reviewer 1 Report
The manuscript proposes a hybrid technique for prediction of a stock trend. This technique is based on the extreme learning machine (ELM) and the discrete wavelet transform approach. The Authors refer to their previous paper and I understand that the present paper is the continuation of the research. The overall impression of the paper is rather good. Although the contribution of this research is in fact only empirical, the findings could be interesting for investors. However, I have several remarks summarized here after.
Major remarks:
- Unfortunately, there is no theoretical contribution of this study because only the existing methods are used. The proposed hybrid DELM ‘model’ is not a (theoretical) model, but just a combination of the existing empirical techniques. Therefore, in my opinion, the Authors should replace the word ‘model’ with the words ‘technique’ or ‘method’ within the whole paper.
- As the paper proposes a hybrid technique (not a model) for prediction of a stock trend, the title is rather confusing and, in my opinion, it should be improved. Maybe it could be:
‘A Hybrid Technique Based on Extreme Learning Machine and Wavelet Transform Denoising for Stock Prediction’
or
‘A Hybrid Method Based on Extreme Learning Machine and Wavelet Transform Denoising for Stock Prediction’
- The Authors state on page 8 that they set the wavelet function as db8. However, for instance, MATLAB offers more than 100 various wavelet functions. The choice of this wavelet function is not clearly and sufficiently justified.
- Moreover, due to my experience, the main disadvantage of the wavelet-based applications in financial market research is that there are a lot of possible wavelet functions, and they should be fitted to each time series, separately. In my opinion, this is the main disadvantage of the proposed method because the Authors use only one db8 wavelet function (without any justification).
Minor remarks:
- The Abstract is too long. It contains many technical and useless information, for instance the values of statistical metrics.
- Moreover, future research directions could be highlighted in the Conclusion section.
- I do not find citations supporting the equations in Subsection 2.1 (ELM).
- The section References must be improved (i.e. authors’ names, titles, journals’ names, pages, etc.). Moreover, DOI numbers are absent, and it is difficult to find and check the references now.
Author Response
Please see the doc file.

Reviewer 2 Report
General
In this paper, the authors explore discrete wavelet transform denoising (DWT) to denoise stock data. Additionally, his paper proposed combining ELM and DWT-based denoising to predict the trend of several stock indices. The authors suggest different precision indicators to test their methods in predicting the trend of stocks, and compare with different machine learning algorithms. The results are convincing and the paper is well-written.
Main comments
Here is a suggestion in which the paper could be improved.
I think authors should consider not referring to their previous works or results in the abstract. “We previously proposed a continuous (…)” But, where? "In our previous research (…)”. I politely suggest that they should cite or compare previous results in the main text of the paper. In this way, they can rewrite the abstract, focusing on their novel and interesting results only.
Besides that, I suggest publication of this work in Entropy in its present form.
Recommendation
In my opinion, the scientific content of the paper is sound and the experimental results are convincing. The treatment of literature seems fair. Furthermore, I think that enough methodological details are provided for this work to be reproduced. Therefore, I recommend that this paper should be accepted for publication after minor revision.
Author Response
Please see the doc file.

Reviewer 3 Report
I would like to see some additions:
1. the paper does not discuss the efficient market hypothesis at all. Stock price predictions, however, contradict it. This should be discussed.
2. the Extreme Learning Machine could be explained more illustratively.
3. please insert text (at least 2 sentences) between main and subheadings, e.g., between 3. and 3.1.
4. the paper often says "In the previous research". This is a bit annoying, because you can't understand the paper correctly if you haven't read the previous research. Please explain the important points in this paper again.
Author Response
Please see the doc file.

Round 2
Reviewer 1 Report
Dear Authors,
According to the new entropy-template:
‘Include the digital object identifier (DOI) for all references where available.’
DOI numbers are available here:
https://www.crossref.org/guestquery/